# Analysing interventions designed to reduce tuberculosis-related stigma: A scoping review

Isabel Foster[1,2], Michelle Galloway[1], Wieda Human[1], Michaile Anthony[3], Hanlie Myburgh[3], Nosivuyile Vanqa[3], Dillon T. Wademan[3], Goodman Makanda[1], Phumeza Tisile[1], Ingrid Schoeman[1], Graeme Hoddinott[3], Ruvandhi R. Nathavitharana[1,4]*

1 TB Proof, Cape Town, South Africa, 2 Global Health, International Development Research Centre, Ottawa, Canada, 3 Desmond Tutu TB Centre, Department of Paediatrics and Child Health, Faculty of Medicine and Health Sciences, Stellenbosch University, Stellenbosch, South Africa, 4 Beth Israel Deaconess Medical Center and Harvard Medical School, Boston, Massachusetts, United States of America

* rnathavi@bidmc.harvard.edu

**Data Availability Statement:** All individual studies analyzed in this scoping review are publicly available. We have provided our study data

## Abstract

Stigma is a critical barrier for TB care delivery; yet data on stigma reduction interventions is limited. This review maps the available literature on TB stigma reduction interventions, using the Health Stigma and Discrimination framework and an implementation analysis to identify research gaps and inform intervention design. Using search terms for TB and stigma, we systematically searched PubMed, EMBASE and Web of Science. Two independent reviewers screened all abstracts, full-texts, extracted data, conducted a quality assessment, and assessed implementation. Results were categorized by socio-ecological level, then sub-categorized by the stigma driver or manifestation targeted. After screening 1865 articles, we extracted data from nine. Three studies were implemented at the individual and interpersonal level using a combination of TB clubs and interpersonal support to target internal and anticipated stigma among persons with TB. Two studies were implemented at the interpersonal level using counselling or a video based informational tool delivered to households to reduce stigma drivers and manifestations. Three studies were implemented at the organizational level, targeting drivers of stigma among healthcare workers (HW) and enacted stigma among HWs. One study was implemented at the community level using an educational campaign for community members. Stakeholder consultation emphasized the importance of policy level interventions and education on the universality of risk to destigmatize TB. Review findings suggest that internal and anticipated TB stigma may be addressed effectively with interventions targeted towards individuals using counselling or support groups. In contrast, enacted TB stigma may be better addressed with information-based interventions implemented at the organizational or community level. Policy level interventions were absent but identified as critical by stakeholders. Implementation barriers included the lack of high-quality training and integration with mental health services. Three key gaps must be addressed in future research: consistent stigma definitions, standardized stigma measurement, and measurement of implementation outcomes.

extraction form as a supplementary file in the Appendix.

**Funding:** This work was funded by a Stop TB Partnership Challenge Facility for Civil Society Round 9 grant, awarded to TB Proof. IF is supported by an International Development Research Centre Scholar Award. RRN is supported by National Institutes of Health Career Development Award (NIAID K23 AI132648-04) and an American Society of Tropical Medicine and Hygiene Burroughs Wellcome Fellowship. The funders had no role in study design, data collection and analysis, decision to publish, or preparation of the manuscript.

**Competing interests:** The authors declare that they have no competing interests.

## Introduction

Globally, tuberculosis (TB) remains a leading cause of death from an infectious disease, with 1.5 million deaths recorded in 2020 (an increase from 2019 due to the impact of the COVID pandemic on TB care) [1]. Stigma, described in 1963 by Goffman as a 'situation of the individual who is disqualified from full social acceptance,' [2] is noted as a significant social determinant of health [3]. Stigma is recognized as a major barrier to the delivery of quality TB care [4–6]. Addressing TB stigma was identified by global leaders as an urgent item to address during the United Nations High Level Meeting on TB [7]. However, there is a paucity of data on evidence based interventions to address TB stigma [8].

Manifestations of stigma (discussed here in the context of TB) can be classified as follows: (i) enacted stigma meaning expressions of stigma such as discrimination, isolation, or judgement from others [9]; (ii) anticipated stigma whereby persons fear how others will act toward them once they come to know their TB status [10]; and (iii) internal stigma, or the process by which persons with TB internalize or endorse negative stereotypes and therefore think or behave accordingly [11]. To reduce health-related stigma, robust interventions are needed, both before and after stigma has been associated with a disease. Interrupting stigma before it has been applied requires targeting its drivers, including misunderstanding of the disease and blame attributed to those infected [12]. Intervening after stigma has been linked to a disease involves addressing the manifestations of stigma and their subsequent manifestations such as community or workplace discrimination, in order to mitigate harm and to shift negative attitudes and behaviours [12]. Clarification regarding the target for intervention on the 'driver-to-manifestation' chain is essential to understand which interventions work and why [13].

The objective for this scoping review was to map the available literature on TB stigma reduction interventions in terms of the manifestations of stigmas targeted by each intervention, as well as the approach to intervention implementation and evaluation. This review applies the Health Stigma and Discrimination Framework, which is based on the principle that stigma is a social construct that requires research, programming and policy efforts, rather than overly focusing on individuals affected by stigma to be the sole target of interventions [12]. As such, we sought to categorize and evaluate interventions at the various socio-ecological levels, namely individual, interpersonal, organizational, community, and policy, at which stigma operates [14].

## Methods

The methodology for this scoping review follows the five phases of the Arksey and O'Malley framework and the Joanna Briggs Institute 2015 recommendations [15, 16]. We followed the preferred reporting items for systematic reviews and meta-analyses extension for scoping reviews (PRISMA-ScR) checklist (see S2 Appendix) to report our study findings [17].

### Phase 1: Defining study scope and research questions

We sought to understand the design and impact of interventions aimed to reduce TB related stigma. We obtained input from stakeholders with lived experiences of TB as well as clinician researchers to formulate this question and to assist with identifying relevant research and operational examples.

### Phase 2: Identifying relevant studies

An initial search of PubMed was undertaken followed by an analysis of the text words contained in the title and abstract of relevant articles, and of the index MESH terms used to

describe the articles. Based on the initial search, we refined our search terms for TB and stigma using identified keywords and index terms (S1 Appendix) and undertook a revised search across the following databases (PubMed, EMBASE, Web of Science). All searches were conducted by a medical librarian and results imported into Covidence software (Fig 1) [18].

### Phase 3: Study selection

We considered any studies that implemented interventions to reduce the stigmatization of people with any form of TB (including drug resistant TB). We did not restrict by study design and included studies reporting on quantitative, qualitative or mixed data. We included published studies that assessed interventions that included reducing stigma alongside other clinical outcomes, and studies that assessed impact on stigma alone. We restricted to studies published in English without restriction by publication year.

Articles were screened for relevance based on the title and abstract. This was followed by a full-text review of articles identified from initial screening. At both stages of the screening process, articles were screened by a review group, with decisions regarding article inclusion made by two independent reviewers (IF, MG, WH, DW, HM). Disagreements were resolved through discussion with a third reviewer (RRN).

### Phase 4: Data charting, quality appraisal and stakeholder consultation

Data was extracted from the included papers using a pre-piloted charting table. The charting table incorporated data on study authors and year of publication, country, health system context, study design, target population, sample size, details about how and when the intervention was delivered, analysis methods, primary study findings, and the impact of the intervention. Two reviewers (IF and MG) extracted data independently using NVivo software to organize the data. Any disagreements that arose between the reviewers were resolved through discussion with a third reviewer (RRN). The quality rating of all studies included was graded by two reviewers (IF and WH) using a modified version of the 2018 mixed-method quality appraisal tool (MMAT [19]). Upon answering screening questions for each study type (qualitative, quantitative, and non-randomized control), a total percentage was calculated. Studies which scored less than 50% were of low quality, between 51% and 70% average quality and above 70% high quality. An assessment of implementation strategy and outcomes was conducted using guidance from Proctor et al. [20].

To build on and contextualise our findings for discussion, we undertook a stakeholder consultation, as recommended in the Arksey and O'Malley framework [15], with TB survivors, advocates, policy makers and clinician researchers. Using a focus group format to facilitate knowledge exchange and group feedback, we shared preliminary findings of the review and asked them to discuss issues related to interventions to reduce stigma drawing on their expertise and experiences and to identify emerging priorities for research for such interventions.

### Phase 5: Collation and summary of findings

Three reviewers (IF, MG and RRN) classified interventions aimed to reduce stigma by the socio-ecological level at which the intervention was targeted (individual, interpersonal, organizational, community, or policy level) [14]. Individual interventions targeted individual behaviour change among those with diagnosed with TB to modify individual characteristics such as knowledge, attitudes, and self-esteem. Interpersonal interventions aimed to modify the person living with TB's environment such as their household, work environment and/or friendship networks by establishing or strengthening interpersonal relationships to restore or promote the person with TB's health. Organizational interventions targeted organizational change to

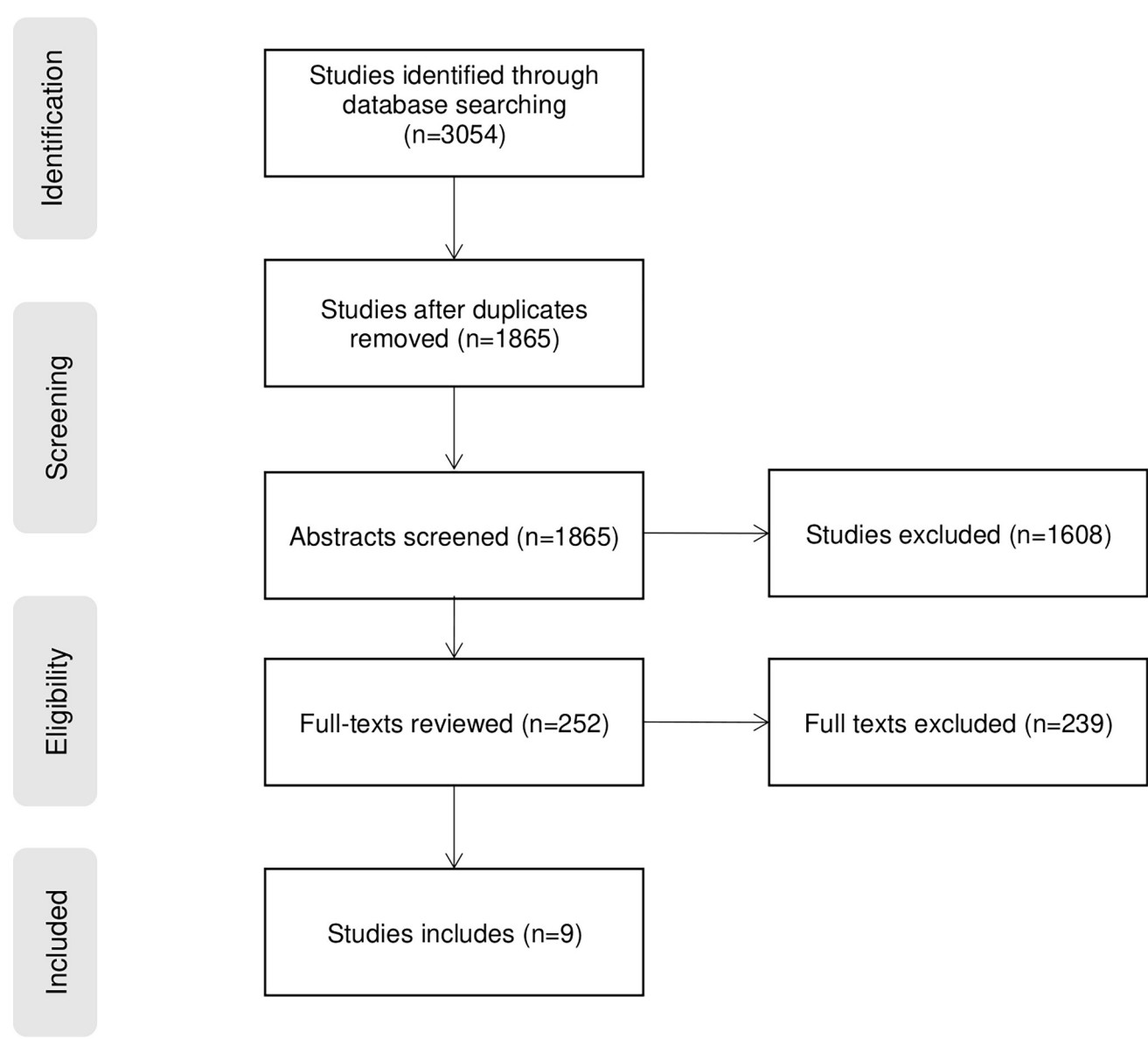

**Fig 1. PRISMA diagram illustrating study selection process.**

modify health and stigma-related aspects of an organization, such as through training programs or organizational policy. Community interventions aimed to increase knowledge related to TB and regarding stigma in specific community groups, and to increase community development skills, such as support network to facilitate access to services for people with TB. Public policy level interventions use national and local laws and policies to effect change [14] (Fig 2). We used recently developed guidance on TB stigma measurement designed with expert consensus input [21] to standardize the definitions used by the reviewers to classify stigma manifestations, rather than definitions used in individual studies to enable comparison across studies. For example, in one study the phrase 'I stay away from people. . . to avoid being rejected' had been classified as internal stigma [22]. However, as per the stigma manifestation classification used in this review [21] this indicator was reclassified as a measure of anticipated stigma.

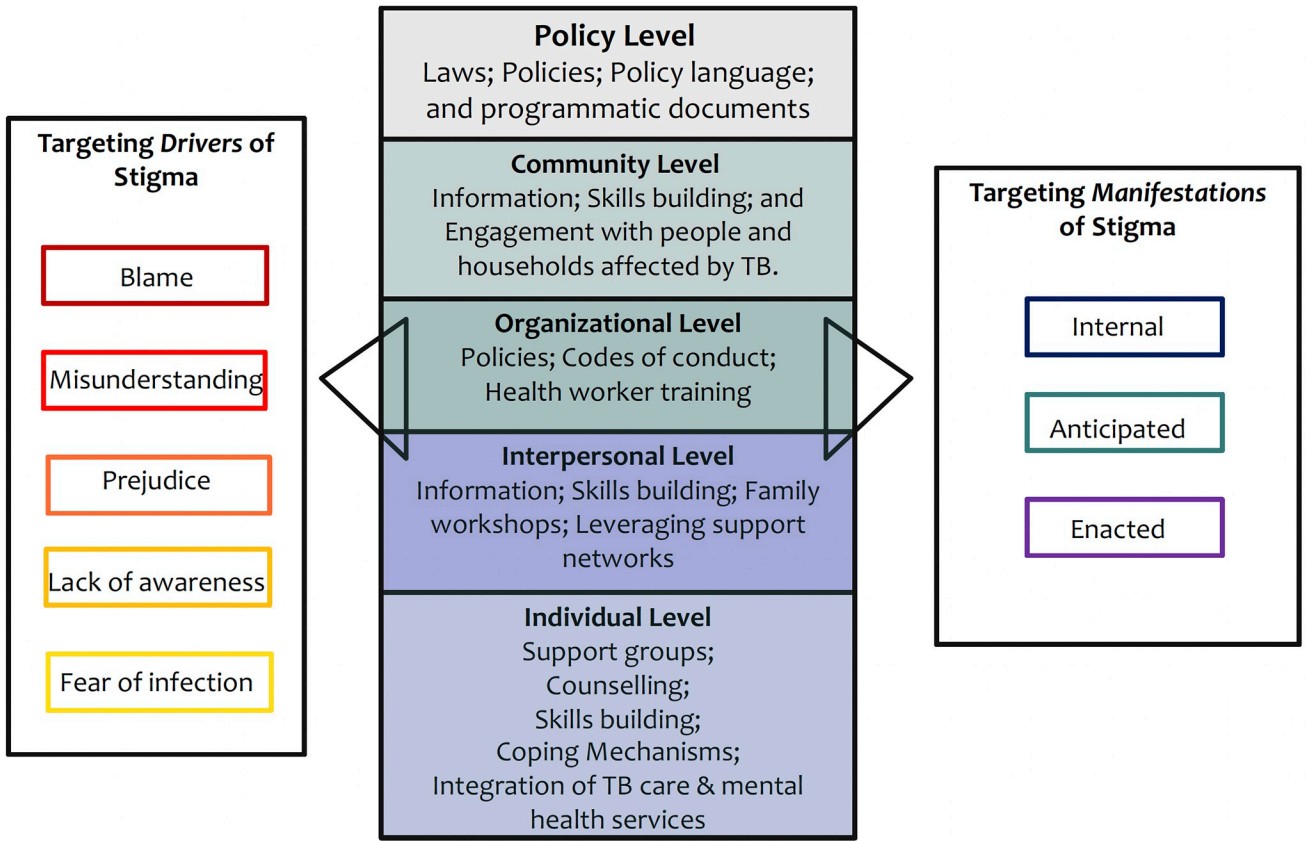

**Fig 2. Potential stigma reduction interventions targeting stigma drivers and/or manifestations that could be implemented at different socio-ecological levels.**

## Results

The database search identified 3054 articles for screening, of which 1189 were duplicates. Of the remaining 1865 abstracts, 1608 were excluded. Of the 252 full texts assessed for eligibility, a total of 239 studies were excluded due to a) not being about a TB stigma intervention evaluation (n = 233), b) not being about stigma (n = 5) or c) not being about TB (n = 1). Overall, nine studies were included (Fig 1). Results are categorized first by socio-ecological level, then sub-categorized by the type of stigma manifestation (internal, anticipated, or enacted) or driver (such as blame or misinformation) that was targeted.

## Characteristics of included studies

Studies focused on the following populations: people with drug-sensitive TB (n = 4), people with multidrug-resistant (MDR)-TB (n = 1), health care workers (n = 3) and community members (n = 1). All studies were conducted in the pilot stage of implementation. Studies focusing specifically on TB patients and their families were implemented at either the diagnosis (n = 1) or the treatment (n = 4) stage of the TB care cascade. Interventions are classified according to the socio-ecological level at which they were targeted as follows: (1) individual or interpersonal (n = 3), (2) interpersonal (n = 2), organizational (n = 3), and community (n = 1). All studies were conducted in lower- and middle-income countries. The characteristics of included studies and findings related to stigma are in Table 1.

**Table 1.  Characteristics of included studies, organized by socio-ecological level.**

| Study Details | | Intervention | | |
| --- | --- | --- | --- | --- |
| Author, Year, Journal | Population, Country | Study Design, Intervention description, Measurement approach | Stigma manifestations or drivers targeted and measurement approach | Stigma and Clinical Outcomes |
| *Individual and Interpersonal Level* | | | | |
| Acha, 2007 Global Public Health | 285 MDR-TB patients Peru | Design: case history evaluating psychosocial support group intervention delivered over a 5 year period (8 groups, average of 6 sessions/group). Intervention: Four components: weekly (and subsequently bimonthly) support groups, recreational excursions, symbolic celebrations, and family workshops. Actors/training: Psychiatrists (who received training in psychosocial support groups) and nurses co-facilitated the sessions, other HWs encouraged participation. | Internal and anticipated stigma Measurement: Participant observation, informal FGDs, session summaries, transcripts and field notes. No stigma measurement tool used. | Stigma outcomes: support groups had positive outcomes on stigma, measured by participants statements. Clinical outcomes: 60% cured, 1% relapsed, 3% died, 4% were lost to follow up (did not complete treatment), and 32% remained in treatment. |
| Demissie, 2003 Social science & medicine | 128 TB patients (64 control, 64 intervention) Ethiopia | Design: Mixed methods cohort study to assess the effectiveness of TB clubs in improving treatment adherence and societal changes in attitudes towards TB. Intervention: TB clubs met weekly in churches, mosques, market places or other venues for social events with 3–10 members and one elected leader. Actors/training: TB club leaders (elected by members) were provided with educational material on TB prepared by the MOH and the Regional Health Bureau. TB leaders also sought help from influential community members like priests to support patients. Local HWs including CHWs supervised the TB clubs. | Stigma drivers: attitudes towards TB Measurement: Focus group discussions and in-depth interviews. No stigma measurement tool used. | Stigma outcomes: Qualitative data suggested improvement in building positive attitudes and reducing misconceptions about TB Clinical outcome: Completion of treatment was higher in the TB club group 56/64 (87.5%) compared to 38/64 (59.4%) in the comparison group. |
| Macq, 2008 BMC Public Health | 268 TB patients (146 control, 122 intervention) Nicaragua | Design: Non-randomized intervention study to evaluate changes in internalized stigma and treatment. Intervention: Patients in intervention municipalities received self-help groups supported by home visits to identify strength and weaknesses of support networks, and to tailor activities to support the patient's needs compared to control municipalities. Actors/training: Interventions were designed in collaboration with the MOH. Training on self-esteem for patients was organized by a psychologist in each of the municipality and then a draft of the interventions package was discussed in each municipality. Municipal nursing staff facilitated the TB clubs. | Internal and anticipated stigma Measurement: Structured Questionnaire using Boyd Ritsher's scale for mental illness (30). | Stigma outcomes: 27.4% of patients agreed to statements indicative of internal and anticipated stigma, compared to 33.1% in the control group. Clinical outcomes: 93% of patients completed treatment compared to 90% in the control group. |
| *Interpersonal Level* | | | | |

*(Continued)*

**Table 1.** (Continued)

| Study Details | | Intervention | | |
|---|---|---|---|---|
| Author, Year, Journal | Population, Country | Study Design, Intervention description, Measurement approach | Stigma manifestations or drivers targeted and measurement approach | Stigma and Clinical Outcomes |
| Bond, 2017 International Journal of Tuberculosis and Lung Disease | 1826 TB patients and 1235 adult members of their households Zambia and South Africa | Design: RCT to evaluate a household intervention aimed at reducing TB transmission and prevalence, not specifically designed to reduce stigma. Intervention: TB screening and HIV counselling and testing for all household members. Actors/training: Household counselors. | Drivers of stigma including blame and judgement Internal and anticipated stigma Measurement: A piloted and adapted TB stigma indicator set with 14 items (4 targeting household members; 10 targeting people with TB). | Stigma outcomes: Internalized stigma was less prevalent in the household arm at both baseline and follow-up, with an adjusted prevalence rate of 0.85 (95%CI 0.41–1.76). Variability in stigma levels post-intervention between countries and across communities was large. |
| Wilson, 2016 J Clin Tuberc Other Mycobact | 1916 TB patients and households El Salvador | Design: Observational pilot study to evaluate feasibility of videography based health promotion strategy to improve patient and family understanding of TB and efficiency of medical evaluation and contact investigation. Intervention: A 7-minute videography-based educational tool utilizing visual aids and patient testimonials to discuss basic principles of TB, including how it can be successfully treated and cured shown to patients and family members upon arrival and return to TB clinics. Actors/training: The video was shown to patients by clinic staff. The video script was composed in partnership with Salvadoran TB clinic physician leaders and nurses and was formally approved by the El Salvador MOH TB program health directors. | Drivers of stigma: misconceptions Measurement: Informal interviews. Specific stigma measurement tool not included. | Stigma outcomes: Nurses perceived a sustainable decrease in the negative stigma and discrimination reported by patients. Clinical outcomes: Nurses reported an approximated 50% decrease in patients subsequently not completing therapy |
| *Organizational Level* | | | | |
| Yassi, 2019 Journal of applied arts & health | 78 HWs South Africa Mozambique Zimbabwe | Design: An embodied multi-country exercise using participatory theatre workshops to raise HW awareness about HIV and TB stigma in the workplace. Intervention: HWs underwent a participatory theatre workshop to document and better understand the short- and longer-term impacts of this alternative training technique on raising awareness about stigma in the workplace and the ways in which it might be overcome as part of improving occupational health. Actors/training: Facilitator from the study group, training not specified. | Drivers of stigma such as feeling lonely, anxious or judged. Enacted stigma Measurement: Exit evaluation, follow-up questionnaires completed six months to one year later, and in-depth discussions two years later. Specific stigma measurement tool not included. | Stigma outcomes: Several HWs mentioned an increased willingness to report incidents of violence, harassment and discrimination and being mindful and respectful of those who did report. Personal behaviors towards colleagues were reported to have also changed as a result of the exercises. |

(*Continued*)

**Table 1.** (Continued)

| Study Details | | Intervention | | |
|---|---|---|---|---|
| Author, Year, Journal | Population, Country | Study Design, Intervention description, Measurement approach | Stigma manifestations or drivers targeted and measurement approach | Stigma and Clinical Outcomes |
| Siegel, 2015 Glob Public Health | 11 HWs South Africa | Design: Pilot studies (to inform trial development) to assess the impact of images that aimed to promote the reduction of HIV and occupational TB stigma and to assure staff of the confidentiality of occupational health units. Intervention: A stigma reduction campaign consisting of visual messaging images were designed to focus on the following attributes: aesthetics, cultural appeal and appropriateness, literacy and 'Ubuntu' (community cohesion and togetherness). Actors/training: Messaging developed was piloted and modified based on input from 11 HWs at 5 hospitals. | Drivers of stigma such as lack of confidentiality. Measurement: Structured interview. Specific stigma measurement tool not included. | Stigma outcomes: Respondents appreciated the explicit association of the office files behind a lock, commenting that it served as a visual reminder of the importance of confidentiality in the workplace. |
| Wu, 2009 J Formos Med Assoc | 1279 HWs Taiwan | Design: Pre/post study implementing training workshops to improve knowledge and reduce stigma in first line HWs. Intervention: Nationwide TB training workshops focusing on TB education, current TB epidemiology in the country, skills required for DOTS execution, and de-stigmatization and human rights. Actors/training: Information not provided. | Type of stigma manifestation not specified. Measurement: 8 questions on stigmatization adapted from Attribution Questionnaire—Short Form—8 Items (AQ-S8) for Measures of Illness Stigma | Stigma outcomes: Pair comparison of stigmatization scores revealed a reduction in stigmatization, with the DOTS workers attaching less stigma to TB patients. |
| | | *Community Level* | | |
| Balogun, 2014 The American journal of tropical medicine and hygiene | 252 Community members Nigeria | Design: Pre/post study to determine the effect of a community based intervention on improving TB knowledge, attitudes and preventive practices among adults in a peri-urban community. Intervention: CVs organized two group health education activities within each month of the intervention phase (3 months) and had individual daily discussions with at least five community members each. CVs organized health talks in the market, church, and mosque and targeted households and different occupational groups. Actors/training: CVs were nominated by the community development and CVs were trained to sensitize the community on TB, identify people at risk of TB, and link people being evaluated for TB to the DOTS clinic. | Drivers of stigma: attitudes and misconceptions Measurement: Structured questionnaire. Specific stigma measurement tool not included. | Stigma outcomes: Less respondents (19%) post-intervention responded that they would feel compassion and a desire to help people with TB people with TB than those interviewed pre-intervention. Clinical outcomes: 8 out of 12 referred patients attended the DOTS clinic of whom one tested positive for TB. |

Abbreviations: MDR: multidrug-resistant, HW: health worker, DOTS: directly observed therapy, short-course, CVs: community volunteers

## Quality assessment

All included articles underwent methodological quality appraisal using the MMAT. One study that was a randomized controlled trial (although stigma was not a primary outcome) scored 100%. However, the majority of studies (range 28–100%) were judged to be low quality, often due to lack of coherence between qualitative data sources, analysis and interpretation and high risk of non-response bias in quantitative studies. No studies were excluded based on quality.

## Stigma measurement

There was considerable heterogeneity in the approaches to TB stigma measurement (Table 1). Four studies used quantitative indicators to compare changes in stigma pre and post intervention [22–25]. Of these studies, two differentiated indicators based on stigma manifestations [22, 25], however these studies classified these indicators into stigma manifestations that were not aligned with the definitions of the stigma manifestations used in this review [13] so these were reclassified. Three studies gave no definition of stigma nor the drivers or domain(s) that were being measured [23–25].

The remaining five studies used qualitative data [26–30], four of which did not incorporate any specific analysis approach or framework and focused on the use of quotes. One qualitative study conducted thematic analysis [27].

## Implementation details evaluated

Detailed descriptions of interventions and indicators of implementation effectiveness (distinct from clinical effectiveness) should be included to enable comprehensive evaluation of interventional studies [20, 31, 32]. We assessed the included studies using Proctor et al.'s recommendations for naming, defining and operationalizing implementation strategies across seven categories: actor, action, action targets, temporality, dose, implementation outcomes addressed and theoretical justification [31] (S3 Appendix). Assessment of implementation outcomes included the following measures: acceptability, adoption, appropriateness, cost, feasibility, fidelity, penetration and sustainability [20]. All studies were implemented in the pilot study phase. No studies adopted any formal theoretical frameworks that are used for implementation research [33]. Appropriateness and feasibility were the most frequently reported implementation outcomes (S3 Appendix). Studies varied in the degree of description of the intervention. Some included intervention descriptions that were sparse and lacked key details that are needed for systematic evaluation and replication [23, 25, 29, 34]. Few interventions documented how they were tailored to different contexts. Those that did, conducted an operational analysis and tailored interventions based on findings and the judgement of local health teams [22]. One intervention described modifications during study implementation for example, when nurses were used to mitigate a shortage of psychiatrists to conduct support groups in Peru [26]. Although only one intervention was at the community-level and used community volunteers as part of the implementation team [23], several others used community actors such as household counselors [25], or CHWs to deliver the intervention and local leaders to provide support [27].

## Combined individual and interpersonal level interventions

No included studies focused solely on targeting stigma experienced at the individual level. Three studies involved interventional components that targeted both individual behavior and behavior of familial and/or social networks of participants. This was done with the aim of reducing anticipated or internal stigma. All three interventions used support groups (often

termed TB clubs consisting of people with TB and a peer or clinician facilitator) [22, 26, 27] at the individual level to equip TB patients with mechanisms to counteract negative beliefs. Similarly, all three interventions also incorporated interpersonal components to the intervention including home visits [22], family workshops [26], and additional support from community leaders [27]. Stigma reduction was the primary intervention aim in only one identified study [22]. The other three studies primarily aimed to assess the impact of an intervention on clinical outcomes: adherence [26] and treatment completion [26, 27] with impact on stigma measured as a secondary outcome.

**Effect on internal stigma.** Two studies reported success in reducing internal stigma. Both used a support group termed either a TB club or psychological support group, as part of a package of interventions that included home visits to identify strengths and weaknesses of support systems in Nicaragua [22] or a combination of recreational excursions, symbolic celebrations, and family workshops for people with MDR-TB in Peru [26]. One study demonstrated a reduction in internal stigma after 2 months, measured using an adapted Boyd Ritsher scale, with an average mean stigma scale score of 27.4 in the intervention group compared to 33.1 in the control group (p = 0.001). The other study reported that qualitative data from participant observation and focus groups suggested a reduction in internal stigma (Box 1), and that cured patients helped newly diagnosed patients reckon with feelings of 'uselessness', such as due to reduced earning capacity caused by side-effects, by showing current patients that these limitations were temporary.

---

### Box 1. Instructive quotes from interventions reducing internal stigma

*Pre-intervention*:

More than half (52.2%) of the initial cohort of patients in treatment for MDR-TB in Lima was diagnosed with a depressive syndrome at baseline [26].

Participants discussed the different emotions that they associated with stigma, including feeling 'lonely', 'anxious', 'unwanted' and 'judged' [30].

*Post-intervention*:

"I would ask God, 'Why do you punish me, so?' But now I realize that it wasn't a punishment because I met a group of real friends, ones that I could truly consider friends and tell them what I'm feeling. And now I'm cured."–TB club participant [26].

Participants recalled the impact that it had on their understanding of stigma and discrimination and its ability to help participants tangibly embody being a victim of such events: '[the exercise is] something that will forever be in someone's mind. [. . .] I know I was stigmatized. The person will never forget [that] okay. This is stigma. This is how a stigmatized person feels' [30].

---

**Effect on anticipated stigma.** In the study evaluating group support for MDR-TB patients in Peru, individuals who reported previously feeling the need to hide side-effects of treatment to mitigate inadvertent disclosure of disease to others subsequently no longer felt the need to do so [26]. The impact of a TB club intervention on anticipated stigma was also qualitatively measured in Ethiopia [27]. This study found that patients who had previously struggled to

disclose positive TB results to family members were able to talk publicly about their disease after participating in weekly TB clubs led by a TB leader elected by the group, who also sought additional assistance from community leaders such as priests to provide patient encouragement and support (Box 2).

---

Box 2. Instructive quotes from interventions reducing anticipated stigma

*Pre-intervention*:

"I've had to learn to be a liar. When people ask why my skin is so dark, I say I've gone to the beach, or into the mountains. I mean, what can I do? I have to come up with whatever lie I can besides the truth, anything but the truth, because if they found out the truth, they would discriminate against me. Who wants to be marginalized? One has to lie." [26]

They said that the diagnosis of TB in the past was a shock for all of them since they may lose their job (e.g. priest, civil servants), be divorced and change their social involvement [27].

"We were all afraid of it. It means that you are going to be weak and not able to support your family. It also means you might lose your social relation and above all it means you are going to die."–TB club participant [27].

TB is still considered a shameful disease; an attitude that has a tendency to cause TB patients to hide their disease and avoid telling others about it [24]

*Post-intervention*:

For many patients the support groups appeared to be one of the only places in which patients could speak freely about their experiences and feelings, and receive empathetic support from others [26].

"It is the second time I am diagnosed. I was hiding my illness. However, I became weak and couldn't farm my land and support my family. Therefore, I was forced to talk to a friend who was member of the clubs. God didn't want me to die. I was ashamed in the beginning but I told my friend and he brought me to the health centre. I am now healthy and a good teacher to the others" [27].

Though targeted conversations with family members, health providers from five TB clinics reported a perceived reduction in both the fear of negative family stigma about TB [28]

---

### Interpersonal level interventions

Two studies intervened at the interpersonal levels of the affected person's household or social networks with the aim of reducing stigma experienced by those living with TB by addressing drivers of stigma as well as internal and enacted stigma manifestations. This included reducing blame associated with the disease [25, 28] and misunderstanding about the biomedical causes of the disease [25].

**Effect on internal stigma.** A randomized controlled trial in South Africa and Zambia designed to evaluate the impact of household-based TB screening, which included household

counselling (practical, clinical and emotional support provided during treatment), on TB prevalence also assessed internal stigma as a secondary outcome. Questionnaire results from household members showed the prevalence of blame, a driver of internal stigma, directed toward TB patients decreased in both intervention and control communities [25]. However the prevalence of internal stigma among TB patients fell in both the household intervention and control communities in South Africa and the intervention community in Zambia, but increased in the control communities in Zambia [25]. One potential explanation was the heterogeneity in community sites between Zambia and South Africa in terms of factors such as livelihood options, population movement, population mix, spatial layout, knowledge of TB, and community identity. No significant difference was found between reduction in the anticipation of stigma between the intervention and control communities, although the prevalence of disclosure of TB status increased in the intervention communities in both Zambia and South Africa [25].

**Effect on drivers of stigma.** In El Salvador, a videography-based educational tool provided (1) basic information about TB, (2) common misconceptions and misunderstandings of TB and (3) patient testimonials was delivered in the outpatient clinic setting and in homes by outreach workers to patients and family members post diagnosis and during return visits to improve family understanding of TB and its treatment to reduce stigma and improve contact investigations [28]. Participants who viewed the videos were reported by nursing staff to have shown reduced fear and stigma among family members and neighbors and reported significant improvements in family support of TB patients, although no formal analysis was provided [28].

## Organizational level interventions

Three interventions were implemented at the organizational level, and were focused on health workers. These studies employed a range of information-based tools, including visual messaging, participatory theatre, and training, to address the drivers of stigma within health care settings.

**Effect on drivers of stigma.** The use of an information-based intervention in the form of visual messaging (delivered as posters displayed around Occupational Health Units) was evaluated as part of an intervention targeted towards health workers (HWs) in South Africa to increase the use of occupational health, decrease TB stigma among HWs, and improve HW health and wellbeing [29]. Using structured interviews, this study showed that HWs preferred messages that were inclusive and direct and that they reported feeling reassured by images that reinforced notions of confidentiality with respect to presenting for evaluation for occupational TB [29].

**Effect on enacted stigma.** In a study that used participatory action theatre, South African HWs learnt that TB stigma can 'happen to anyone' and that it is not one's actions that solicits stigmatization [30]. This provoked reflection on their own past discriminating behaviors towards people with TB leading to hypothesized reductions in enacted stigma (Box 3). As a result of this intervention, several HW participants mentioned greater willingness to report incidents of enacted stigma towards patients and requesting to meet with hospital management to find solutions for issues around TB stigma and discrimination in the workplace.

---

Box 3. Instructive quotes from interventions reducing enacted stigma

*Pre-intervention*:

Most patients had experienced, to varying degrees, social rejection and discrimination from family members, friends, neighbours, and/or health providers. [26]

"Instead of encouraging the patient to seek treatment they would keep him/her at home in order not to be identified as a case of TB."–TB club participant [27]

---

*Post-intervention*:

Most participants who have completed treatment have now successfully reintegrated into their social and community lives; they have returned to work, recommenced their studies, and resumed former family roles and responsibilities [26].

"My wife is with me now, this was not the case before. In the past even if she wanted, her relatives would not let her stay with me. Now everyone knows about the disease, it is not transmitted sexually"–family member of TB patient [27]

"It showed us the reality of how we made people feel when we stigmatize them. Sometimes we're doing it deliberately, sometimes not. But at the end of the day the person would [. . .] not come forward to tell us things [. . .]. We are the ones that are feeling that our people don't want to tell us, but we are the ones that are discriminating against them".–HCW [30].

In another study that delivered nation-wide TB training workshops to HWs in Taiwan, knowledge was not found to be significantly correlated with stigmatization (the type of stigma manifestation was unspecified) (pre-test, $p = 0.298$; post-test, $p = 0.821$) [24]. However, HWs who had themselves had TB and HWs involved in providing Directly Observed Therapy (DOTS) (compared to other unspecified public health workers) were significantly less likely to stigmatize patients ($p = 0.031$ and $p = 0.038$).

## Community level interventions

One intervention at the community level aimed to reduce stigma by increasing knowledge through an educational campaign delivered to community members regarding TB disease or reducing stigmatizing behaviors [23].

**Effect on drivers of stigma.** In Nigeria, after an educational campaign delivered by community volunteers, 34.9% of community members surveyed responded that while they would feel compassion towards people with TB, they would stay away from them compared to 13.5% pre-intervention [23]. The authors hypothesized that this may reflect a problem with the quality and/or content of the training received by community volunteers delivering the intervention.

## Barriers to implementation

TB related stigma was found to be highly contextual, as demonstrated after the prevalence of internal stigma after the same household counselling intervention applied to two study sites in South Africa and Zambia was reduced in one setting but increased in the other, although the confidence intervals were wide [25]. This was suggested to be due to heterogeneity in community sites between Zambia and South Africa in terms of local contextual factors. Recruiting and appropriately training intervention facilitators and addressing mental health in people with TB were identified as important barriers across several studies. When psychosocial support groups were used to reduce stigma in Peru, it was challenging to finding trained facilitators that did not show prejudice or fear of MDR-TB patients. However, despite having little or no training, nurses who served as co-therapists with the trained psychiatrists were able to develop therapeutic strategies to facilitate interventions in the groups and to lead a group when one psychiatrist had to withdraw [26]. In the community based study in which enacted stigma increased

after an awareness campaign, the authors hypothesized that the community volunteers who administered both the campaign and survey could have passed on discriminating views of their own to participants [23]. One of the studies implementing support groups targeted towards reducing individual level stigma found that at baseline, some participants had suicidal ideations and/or depression (Box 1) [26]. Analysis of these barriers highlights the need to ensure high-quality training for staff delivering interventions in studies and moreover to programmatic staff delivering TB care, and for integrated mental health services to be available for people with TB, in addition to interventions targeting stigma.

## Discussion

This scoping review, which used a systematic approach to identifying TB stigma intervention studies coupled with insights from a multi-stakeholder consultation, demonstrated a persistent major gap in the literature regarding the implementation and evaluation of TB stigma interventions. None of the nine stigma reduction intervention studies identified, were designed with the singular intention of reducing the impact of stigma for people living with TB. All interventions were in the pilot stage and were heterogeneous in their approach to both implementation and measurement, with most adopting a variety of tactics that targeted stigma at either a combination of the individual and interpersonal level or at the organizational level (Fig 2). No interventions targeted structural or policy level changes, although the adaptation of policies and practices that perpetuate TB stigma has been identified as a core tenet for the elimination of TB stigma [35].

Prior reviews on TB stigma have highlighted the scarcity of published TB stigma assessments and particular need to address internal stigma experienced by people with TB [22], the need to ensure diversity in patient populations assessed for stigma and use of validated survey tools to quantify the impact at different steps in the TB cascade of care [36], and the lack of interventions to address TB related stigma in health facilities [37]. We note that four further studies have been included in this review, compared to the most recent prior review [8], and that these studies include stigma interventions targeted towards health workers rather than focusing only on people with TB. While our findings align with prior reviews regarding the low number and quality of published studies on this topic, we provide additional insights based on applying a socio-ecological framework to categorize interventions, evaluating implementation processes and outcomes, and conducting a stakeholder consultation to contextualise our findings.

Studies included in our review suggest that internal and anticipated TB stigma manifestations may be addressed effectively with interventions targeted towards individuals in the form of counselling or support groups. In contrast, enacted manifestations and drivers of TB stigma may be better addressed with information-based interventions that are implemented at the household or community level. Both internal and enacted HIV stigma have been associated with a higher prevalence of mental health disorders [5], suggesting that ensuring access to quality mental health services may also be critical for people affected by TB experiencing these forms of stigma. Further, in communities with high co-prevalence of HIV and TB, intersectional stigma is common, as noted in several included studies [25, 29, 30], and should be addressed in intervention design. Variability in study communities was identified as a reflection on the complex nature of stigma that requires improved local contextual understanding to design and implement effective interventions. Authors noted that stigma-specific interventions are required to effectively address TB stigma rather than interventions with other primary aims. Important implementation barriers to overcome include the need for high-quality training regarding stigma and/or TB for those implementing interventions that should address

conscious and unconscious biases, since negative attitudes toward people with TB can otherwise be perpetuated and spread by those delivering the intervention. Moreover, intervention designers and implementers should reflect on and aim to address the broader hegemonic dimensions of stigma, which often reflect societal inequities and racism, that mediate the risk of acquiring TB and poor TB outcomes, and also serve as drivers for exacerbated and intersectional stigma [38].

Our assessment of these studies revealed three critical gaps in the existing literature on TB stigma interventions. Firstly, few studies included in the review differentiated between stigma manifestations, and no studies used the same definition of stigma manifestations, which is essential to allow for comparison among studies. As such, we were obliged to re-classify stigma manifestations to align with the definitions used within this review. Secondly, no study used the same method to systematically assess the impact of the intervention reviewed on stigma. Improved reporting of stigma measurement, specifically categorized by manifestation, would enable comparison of findings across study and practice settings and inform future stigma implementation studies. Guidance on more rigorous and standardized measures of stigma manifestations are available [39], although it remains necessary to consider adaptation of stigma measurement tools to ensure that they are context-appropriate and specific, thereby improving study validity. Thirdly, implementation outcomes were sparsely measured. Most studies focused on intervention adoption and feasibility, few assessed acceptability, appropriateness, cost, penetration or sustainability [20]. Such outcomes are essential to report to facilitate uptake in other settings and to justify scale-up of successful interventions that are both acceptable and sustainable.

A comprehensive guide on stigma measurement published by USAID and KNCV and developed in collaboration with the Stop TB Partnership in 2018 offers definitions of stigma manifestations (internal, anticipated, and enacted stigma) as well as detailed recommendations on the approaches to measuring and addressing these different forms of stigma [21]. Applied to HIV, Zelaya et al., 2012 have also shown how stigma can be best be assessed using indicators across stigma manifestations [10]. Stigma evaluations and intervention design should address intersectional stigma, for example, due to HIV coinfections or other marginalized characteristics such as gender or race that can also modulate other behaviours such as substance use [5]. Most interventions, as evaluated in the nine studies in this review, did not differentiate between manifestation, and targeted stigma very generally, limiting their applicability to other contexts. Additionally, different types of stigma are known to affect different stages of the TB care cascade, and some of the included studies directed their interventions as such but with limited nuanced discussion about how interventions were tailored to the patient population or clinical context. Few studies assessed at which point within the cascade of TB care, and which level of the socio-ecological framework, an intervention should be directed towards to reduce specific forms of stigma. Heijnders and Van der Meij (2006) suggest that multi-component stigma-reduction that target a range of actors and socio-ecological levels are more likely to be effective [14].

## Stakeholder consultation

This review was conducted as part of a community-engaged research study to evaluate TB stigma using community-based stigma assessments in South Africa and use these findings to develop stigma reduction interventions. To complement and contextualize the results, a stakeholder consultation was conducted in the form of a group discussion with policy makers, TB programme specialists, former TB survivors and clinical researchers, all with relevant expertise and experience related to TB management in South Africa. Their opinions closely reflected the

findings of the review: more research must be done to highlight at which point(s) of the cascade of care, with whom, and through what implementation mechanisms interventions should be targeted. The lack of interventions at the policy level was identified as a significant research gap. While national documents detailing strategies to end TB stigma exist in countries such as India, stigma remains all too pervasive in practice settings [40]. A TB programme specialist mentioned that the law should be utilized to a greater degree to ensure that the rights of those living with TB are protected from stigma at the organizational and policy level.

Stakeholders noted that despite growing attention of the need to address stigma, misunderstanding of TB and risk of infection among community members remains widespread. They suggested a greater emphasis on highlighting the universality of TB risk to the general public rather than only focusing on high-risk groups such as people living with HIV, to minimize the exacerbation or transfer of additional stigma. Additional actors both within the healthcare system and in the community should also be engaged to reduce stigma, who may include traditional healers and church leaders who often overlook key entry points to the TB cascade of care. Interventions that equip these groups with tools to mitigate stigma drivers may have great potential to reduce stigma manifestations. Stakeholders also reiterated the importance of ensuring access to both counselling and quality mental health services for people affected by TB.

Strengths of our scoping review included conducting a systematic search of TB stigma reduction interventions literature, that was not limited by country setting or date, in consultation with a research librarian, followed by screening, extraction and evidence grading by at least two independent reviewers. By including only peer-reviewed articles, ensuring that our eligibility criteria were rigorously applied, and performing methodological quality appraisal at the individual study using the MMAT tool, we have endeavored for this scoping review to uphold similar quality standards to a systematic review and maximised our ability to evaluate the impact of stigma reduction interventions. Through the stakeholder consultation that we undertook as part of this review, we were able to incorporate key perspectives from people directly affected by TB. Our study was limited by the inconsistent classification of stigma manifestations and the interpretation of these findings. A further limitation of this review is holding the stakeholder consultation only within the South African context. While the TB burden in South Africa is one of the highest globally, findings may not be applicable to other high incidence settings. Further, since we restricted our search to published literature, grey literature of relevance may have been missed in this review. However, through the stakeholder consultation, relevant international policy documents were identified.

## Conclusion

This scoping review identified key gaps in implementation research aimed to reduce TB related stigma. By mapping interventions implemented across the socio-ecological framework, we highlighted which interventions may be best implemented to address drivers versus manifestations of TB-related stigma and discussed barriers to implementation. While qualitative and quantitative data reported by the included studies suggest that certain interventions may reduce TB stigma, the lack of standardization in stigma measurement approaches and overall low quality of included studies make comparison across studies difficult. Standardized tools to define and measure stigma should be used more widely to ensure that context-specific interventions can be assessed and adapted to suit other environments. Addressing TB-related stigma is essential to both improve access to care and quality of care. However, doing so at the scale that is required to address the TB pandemic will require a more nuanced understanding of what works and why, which will be facilitated by high quality implementation science studies and improved engagement with people and communities affected by TB.

## Supporting information

**S1 Appendix. TB stigma search terms.**
(DOCX)

**S2 Appendix. PRISMA scoping review checklist.**
(PDF)

**S3 Appendix. Assessment of implementation strategy and outcomes for included studies.**
(DOCX)

**S4 Appendix. Data extraction form.**
(XLSX)

## Acknowledgments

We are grateful to Megan McNichol and Diane Young with the Beth Israel Deaconess Medical Center Library Services, who helped design and conduct the searches for this scoping review.

## Author Contributions

**Conceptualization:** Isabel Foster, Michelle Galloway, Michaile Anthony, Hanlie Myburgh, Dillon T. Wademan, Goodman Makanda, Phumeza Tisile, Ingrid Schoeman, Graeme Hoddinott, Ruvandhi R. Nathavitharana.

**Formal analysis:** Isabel Foster, Michelle Galloway, Wieda Human, Ruvandhi R. Nathavitharana.

**Funding acquisition:** Ingrid Schoeman.

**Methodology:** Isabel Foster, Michelle Galloway, Wieda Human, Michaile Anthony, Hanlie Myburgh, Nosivuyile Vanqa, Dillon T. Wademan, Ruvandhi R. Nathavitharana.

**Supervision:** Graeme Hoddinott, Ruvandhi R. Nathavitharana.

**Visualization:** Isabel Foster.

**Writing – original draft:** Isabel Foster, Michelle Galloway, Ruvandhi R. Nathavitharana.

**Writing – review & editing:** Michelle Galloway, Wieda Human, Michaile Anthony, Hanlie Myburgh, Nosivuyile Vanqa, Dillon T. Wademan, Goodman Makanda, Phumeza Tisile, Ingrid Schoeman, Graeme Hoddinott, Ruvandhi R. Nathavitharana.

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
