## [Decision Letter · Decision Letter 0]

2 Feb 2022

PGPH-D-21-00947

Analysing interventions designed to reduce tuberculosis-related stigma: a scoping review.

Dear Dr. Nathavitharana,

Thank you for submitting your manuscript to PLOS Global Public Health. After careful consideration, we feel that it has merit but does not fully meet PLOS Global Public Health’s publication criteria as it currently stands. Therefore, we invite you to submit a revised version of the manuscript that addresses the points raised during the review process.

We look forward to receiving your revised manuscript.

Kind regards,

Elizabeth Fair, PhD, MPH

Academic Editor

Journal Requirements:

1. Boxes should be editable and included within the manuscript file like tables. Please amend this.

2. We have noticed that you have uploaded supporting information but you have not included a list of legends.  Please add a full list of legends for all supporting information files (including figures, table and data files) after the references list. 

3. In the online submission form, you indicated that "All individuals studies analyzed in this scoping review are publicly available. The aggregate datasets used and analysed during the current study are available from the corresponding author on reasonable request.". All PLOS journals now require all data underlying the findings described in their manuscript to be freely available to other researchers, either 1. In a public repository, 2. Within the manuscript itself, or 3. Uploaded as supplementary information.

4. Please amend your detailed Financial Disclosure statement. This is published with the article, therefore should be completed in full sentences and contain the exact wording you wish to be published.

iii). State what role the funders took in the study. If the funders had no role in your study, please state: “The funders had no role in study design, data collection and analysis, decision to publish, or preparation of the manuscript.”

Additional Editor Comments (if provided):

Dear Dr. Nathavitharana,

Thank you for submitting your manuscript to PLOS Global Public Health. After careful consideration, we feel that it has merit and has the potential to be an important contribution to the current discourse on stigma and TB, but does not fully meet PLOS Global Public Health’s publication criteria as it currently stands. Therefore, we invite you to submit a revised version of the manuscript that addresses the points raised during the review process.

Please review the detailed comments and contact us if you have any questions.

Reviewers' comments:

Reviewer's Responses to Questions

**Comments to the Author**

1. Does this manuscript meet PLOS Global Public Health’s publication criteria? Is the manuscript technically sound, and do the data support the conclusions? The manuscript must describe methodologically and ethically rigorous research with conclusions that are appropriately drawn based on the data presented.

Reviewer #1: Yes

Reviewer #2: Yes

2. Has the statistical analysis been performed appropriately and rigorously?

Reviewer #1: N/A

Reviewer #2: N/A

3. Have the authors made all data underlying the findings in their manuscript fully available (please refer to the Data Availability Statement at the start of the manuscript PDF file)?

Reviewer #1: Yes

Reviewer #2: Yes

4. Is the manuscript presented in an intelligible fashion and written in standard English?

Reviewer #1: Yes

Reviewer #2: Yes

5. Review Comments to the Author

Reviewer #1: This is a very well written review. Findings of this review is in sync with the systematic review by Sommerland et al on the same topic. Authors have highlighted the significance of having a standard definition and measurement of stigma for better implementation.

Since the authors used their definitions to assess and categorize the stigma measurement indicator across studies, there is the possibility of misclassification. It would be good to explain how this was taken care of.

Further it would be prudent if there was a mention of the possible limitations that the study might have had.

Since the search was restricted to peer reviewed articles there is the possibility of having missed relevant literature from certain low and middle income countries with significant TB burden. Perhaps this should be looked into

Reviewer #2: A very good narrative synthesis and relevant to current discourses around stigma mitigation, and will add value to the current discourse on stigma and TB. But more explicit application of the stated ecological framework is needed in categorizing the review findings, and results sections needs more organization to enhance clarity and coherence. Some suggestions:

1. Suggest more explicitly stating how this review differs from / builds on prior reviews on the topic of TB stigma (eg, Sommerland, especially as it is the most recent, but also others such as Courtwright & Turner, Macq, Nyblade, etc). I do see Sommerland mentioned (line 391) but given cursory attention. That review went into substantial detail into measurement methods as well. Suggest highlighting any new studies found through the current search and specifying the ‘additional insights’ thereby attained, as suggested at the very end of the paper (lines 462 464).

2. At least one study was an RCT, and worthy of mention in the quality section.

3. The categories of interventions are either not defined or then not ideally utilized. For example, no definitions for “support groups”, information” and “counselling” are provided; these would be crucial and should not be assumed. Also, while the classifications of individual, interpersonal and community are provided, the subsequent allocation of interventions into each does not seem accurate. Support groups are classified as individual level interventions but they have a strong interpersonal element. Indeed, the researchers’ own definition (line 276) applies to support groups (“reinforcing social networks” – a peer network is a valued social network). See next point as well.

4. Having these two axes or classifications for interventions (ie, individual, interpersonal, community and support groups, information based, counselling) is weakening coherence of this otherwise very important paper. The authors would be recommended to consolidate (ie, pick one framework for classification, and subsume the other into it). As the ecological model is already explicitly stated as a desired framework, the former classification of individual-interpersonal-community would work. What is missing is structural or policy level, and should be remarked within the discussion if no papers in this regard were located.

a. Accordingly, classification/categorization of interventions in Table 1 should change to be individual, interpersonal and community.

5. Information in Tables 1 and 2 should be reconciled and consolidated. For example: Table 1 should have columns on measurement tools and approaches. The final column of Table 1 at this time has no information on the construct/measure (eg, anticipated stigma, driver of stigma) that was addressed/changed. By combining measurement information from Table 1 and 2, this crucial detail will automatically become more clear to a reader.

6. Table 3. It is unclear what value it adds to the discussion given that the studies in which those measures were assessed is not shared. The information, if imperative, may be added to Table 1. Otherwise, the table can be moved to a supplementary file.

7. Under “Intervening at the Individual level”, what is the subtitle “stigma manifestations” alluding to? The same goes for the other sub-titles (stigma drivers and manifestations) within subsequent two sections. An introductory statement is needed to tie in the text with each of these subtitles, and thereafter a more authentic narrative synthesis is needed to ascertain what exactly the reader should take from the authors’ review. As an example, rather than listing studies from Peru, Nicaragua and South Africa/Zambia one after the next (Line 241 onward), perhaps use your analytic framework to consolidate the main/overlapping points: Two studies successfully reduced internal stigma. One, from XYZ, utilized [said intervention components]… whereas the other, from ABC, implemented […]. A third study also acting at the individual level, reduced anticipated stigma…

8. The terms domains and manifestations are used to describe the same concept (Fig 2 and results uses manifestations, whereas methods and discussion uses domains). Suggest more consistency.

9. Line 340 – may be clearer to state the intervention was successful in only one setting.

10. In this same section, it would be helpful to have a quick summary of what the main barriers were (even if they were only reported in 2-3 studies). There is a final implication suggested for just one of the barriers (mental health) but not the other barrier (discriminatory attitudes of staff).

11. The stakeholder consultation is commendable but the process and purpose remains unclear. Why was the consultation held in South Africa alone (what are the limits thereof), and how many individuals were engaged? It is unclear if stakeholders’ inputs informed the authors’ review/ narrative synthesis; suggest being explicit about this. If they were engaged, more language about their participation in analysis should be shared. If they were not engaged in analysis but rather served as a final space for exchanging feedback, then the entire consultation and ideas emanating thereof may be better situated within the discussion alone (eg, as a knowledge exchange strategy). To name stakeholders as participants suggests a new research study was undertaken, so suggest using more apt terms to describe the stakeholders involved.

12. Given this consultation, did the authors explore the consultation or engagement of community actors within the 9 studies reviewed? Is there space to comment on their inclusion (or exclusion) in the process of intervention design, implementation and/or evaluation?

13. Several of the studies were undertaken in settings with high TB-HIV co-prevalence where there was likely overlapping stigma. Yet there is no remark about this in the review analysis. It is mentioned by a stakeholder. The authors need to connect the stakeholder consultation to the review findings.

13. Points stated in lines 407-409 are crucial. Is there any reference or connection to such findings from other disease states or research projects that could develop this point further? It would be a missed opportunity is authors do not raise the issue of power, inequity and/or the hegemonic dimensions of stigma, even if briefly.

6. PLOS authors have the option to publish the peer review history of their article (what does this mean?). If published, this will include your full peer review and any attached files.

**Do you want your identity to be public for this peer review?** For information about this choice, including consent withdrawal, please see our Privacy Policy.

Reviewer #1: No

Reviewer #2: No

---

## [Decision Letter · Decision Letter 1]

9 Aug 2022

Analysing interventions designed to reduce tuberculosis-related stigma: a scoping review.

PGPH-D-21-00947R1

Dear Dr. Nathavitharana,

We are pleased to inform you that your manuscript 'Analysing interventions designed to reduce tuberculosis-related stigma: a scoping review.' has been provisionally accepted for publication in PLOS Global Public Health.

Best regards,

Elizabeth Fair, PhD, MPH

Academic Editor

Thank you for your very thorough response to all of the first round of reviewer comments. This is a strong paper on the important topic of TB stigma reduction interventions. There remain some grammatical errors, therefore we recommend a final close read for copy editing.

Reviewer Comments (if any, and for reference):

Reviewer's Responses to Questions

**Comments to the Author**

1. If the authors have adequately addressed your comments raised in a previous round of review and you feel that this manuscript is now acceptable for publication, you may indicate that here to bypass the “Comments to the Author” section, enter your conflict of interest statement in the “Confidential to Editor” section, and submit your "Accept" recommendation.

Reviewer #3: All comments have been addressed

2. Does this manuscript meet PLOS Global Public Health’s publication criteria? Is the manuscript technically sound, and do the data support the conclusions? The manuscript must describe methodologically and ethically rigorous research with conclusions that are appropriately drawn based on the data presented.

Reviewer #3: Yes

3. Has the statistical analysis been performed appropriately and rigorously?

Reviewer #3: N/A

4. Have the authors made all data underlying the findings in their manuscript fully available (please refer to the Data Availability Statement at the start of the manuscript PDF file)?

Reviewer #3: (No Response)

5. Is the manuscript presented in an intelligible fashion and written in standard English?

Reviewer #3: Yes

6. Review Comments to the Author

Reviewer #3: The authors have done a good job of responding to reviewer comments. There are some small grammatical errors throughout the paper, so I would suggest one last through editing by the authors. But otherwise, I would accept the re-submitted manuscript.

7. PLOS authors have the option to publish the peer review history of their article (what does this mean?). If published, this will include your full peer review and any attached files.

**Do you want your identity to be public for this peer review?** For information about this choice, including consent withdrawal, please see our Privacy Policy.

Reviewer #3: No
